# Network Pharmacology and Molecular Dynamics Identified Potential Androgen Receptor-Targeted Metabolites in *Crocus alatavicus*

**DOI:** 10.3390/ijms26083533

**Published:** 2025-04-09

**Authors:** Zhen Ding, Yuanfeng Lu, Jichen Zhao, Daoyuan Zhang, Bei Gao

**Affiliations:** 1State Key Laboratory of Ecological Safety and Sustainable Development in Arid Lands, Xinjiang Institute of Ecology and Geography, Chinese Academy of Sciences, Urumqi 830011, China; dingzhen22@mails.ucas.ac.cn (Z.D.); luyuanfeng@njfu.edu.cn (Y.L.); zhaojichen24@mails.ucas.ac.cn (J.Z.); 2Xinjiang Key Laboratory of Conservation and Utilization of Plant Gene Resources, Xinjiang Institute of Ecology and Geography, Chinese Academy of Sciences, Urumqi 830011, China; 3University of Chinese Academy of Sciences, Beijing 100019, China; 4College of Life Sciences, Nanjing Forestry University, Nanjing 210008, China

**Keywords:** *Crocus*, network pharmacology, molecular docking, prostate cancer

## Abstract

The objective of this study is to identify the active components of *Crocus alatavicus* and potential targets through a combination of network pharmacology, molecular docking technology combined with molecular dynamics simulation, and binding free energy analyses. A total of 253 active ingredients from *C. alatavicus* were screened, and 1360 associated targets were predicted through systematic searches conducted using TCMSP, SwissDrugDesign, and SymMap, which were integrated to construct a pharmacological network to dissect the relationships among active components, targets, diseases, and pathways; we found prostate cancer-related genes were significantly enriched among the targets. Subsequently, the core prostate cancer-related targets were identified in the network, and the binding interactions between protein targets and active components were evaluated using molecular docking technology. Furthermore, molecular dynamics simulation and binding free energy analyses were performed to verify the binding stability of the most promising complex. Then, protein–protein interaction network analysis was conducted to evaluate the core target sites, leading to the identification of nine target proteins with significant correlations, providing potential targets for cancer treatment. Furthermore, these targets were found to be associated with 20 signaling pathways, including neuroactive ligand–receptor interactions, prostate cancer, lipid metabolism and atherosclerosis, as well as calcium signaling pathways. The active component–target–disease–pathway network diagram suggests that Capillarisin, Eugenol, 1-(4-Methoxyphenyl)-1-propanol, 2,4,2′,4′-tetrahydroxy-3′-prenylchalcone, and 4-Hydroxymandelonitrile may serve as key components targeting prostate cancer. Molecular docking analyses demonstrated that Capillarisin has a high affinity for the androgen receptor (AR), and molecular dynamics simulation was performed to further verify the binding stability, indicating that Capillarisin may exert its pharmacological effects in prostate cancer. Based on the integrated strategies of network pharmacology, molecular docking, molecular dynamics simulation, and binding free energy analysis, this study generated novel insights into the active components of *C. alatavicus* and potential targets related to prostate cancer, thus providing valuable biological resources for future drug research and development.

## 1. Introduction

Natural products have historically served as chemical templates for pathway-specific drug discovery, exemplified by paclitaxel and vinblastine—two plant-derived molecules that revolutionized our understanding of mitotic machinery dysregulation in cancer [1,2]. Paclitaxel (isolated from *Taxus brevifolia*) binds β-tubulin’s taxane site, locking microtubules in a hyper-stabilized state. This targets the aberrant mitotic spindle dynamics characteristic of rapidly dividing cells (e.g., increased tubulin turnover in breast cancer cell lines) [1,2]. Vinblastine (from Catharanthus roseus) disrupts microtubule polymerization by binding tubulin dimers, preferentially affecting cells with defective spindle assembly checkpoints—a hallmark of lymphoma and testicular cancer genomes [3,4]. Its development trajectory highlights how natural product scaffolds (e.g., vinca alkaloid skeleton) provide structural starting points for optimizing target engagement (e.g., nanomolar affinity for tubulin). These case studies underscore the resource value of natural products in oncology; they are not “antitumor drugs” per se, but rather discovery tools that reveal actionable mechanisms (e.g., microtubule dynamics as a therapeutic axis). Such mechanistic insights position natural products as upstream contributors to precision oncology—bridging genomic aberrations (e.g., *BRCA2* mutations) with druggable protein states.

*Crocus alatavicus* is a perennial early spring ephemeral plant belonging to the Iridaceae family, primarily distributed in Xinjiang, Northwest China. The corm of this plant is a commonly utilized medicinal resource in Kazakh folk medicine. It is often mashed and consumed with horse milk to treat various ailments, including wind-cold-dampness arthralgia, swelling and pain, as well as weakness in the lower back and extremities [5]. Currently, research on the efficacy of *C. alatavicus* is limited to preliminary investigations focusing on the analgesic and anti-inflammatory effects of the alcohol extract derived from its corm [6]. Considering the significant potential of natural products such as paclitaxel and vinblastine in the realm of antitumor drug development, an in-depth investigation into the chemical constituents and bioactivities of *C. alatavicus*, a plant with well-established traditional medicinal applications, is of particular importance.

Network pharmacology is an emerging discipline grounded in systems biology theory and biological system network analysis, and it represents a promising methodology for the design of multi-target drug molecules through the selection of specific signaling nodes [7]. This approach emphasizes the multi-channel regulation of signaling pathways, enhancing the pathway modulation efficacy of drugs while minimizing side effects. Consequently, network pharmacology aims to improve the success rate of preclinical discovery for new drugs and reduce the overall costs associated with chemical probe discovery [8,9,10,11,12]. Moreover, molecular docking simulation technology and artificial intelligence tools of structure biology will also conspicuously accelerate the processes for drug discovery and development.

Prostate cancer ranks among the most prevalent male malignancies, characterized by deregulated androgen receptor (AR) signaling and DNA repair gene mutations—key molecular hallmarks driving its pathogenesis [13]. Approximately 5.5% of prostate cancer cases harbor germline mutations in DNA repair genes (e.g., *ATM*, *BRCA1/2*), which disrupt genomic stability and promote oncogenesis [14]. For instance, *BRCA2*-mutant tumors exhibit hyperactivation of AR transcriptional programs, a phenotype linked to aggressive disease progression [15]. Current clinical management—including surgery, radiotherapy, and androgen deprivation therapy (ADT)—addresses these hallmarks but is limited by targeted resistance (e.g., AR splice variants in castration-resistant prostate cancer) and severe side effects (e.g., neuropathy, metabolic dysfunction) [16,17]. This highlights an unmet need for novel molecular entities that modulate dysregulated pathways (e.g., AR signaling, DNA repair) with improved specificity.

In this study, the active ingredients of *C. alatavicus* were investigated employing a network pharmacology framework (Figure 1), revealing the mechanisms of action of these components from multi-component, multi-target, and multi-channel perspectives. Our findings explore the interactions between potential active components of *C. alatavicus* and prostate cancer-related targets and offer additional potential chemical probes for AR-driven prostate phenotype pathway modulation. By analyzing the target regulatory mechanisms of natural products, it screens precursor resources for drug discovery and provides candidate molecules and clues for subsequent drug development.

## 2. Result

### 2.1. Active Substance Screening and Target Prediction

The TCMSP (https://old.tcmsp-e.com/tcmsp.ph, accessed on 15 October 2024) was utilized to analyze the oral bioavailability (OB) and drug likeness (DL) from *C. alatavicus*, yielding corresponding target information (Appendix A). The identified substances were screened using SwissDrugDesign (http://www.swissadme.ch/, accessed on 15 October 2024), while their corresponding targets were predicted through SwissTargetPrediction (http://www.swisstargetprediction.ch/, accessed on 15 October 2024). The target information for all detected substances was subsequently compiled and summarized (Appendix A). Additionally, the SymMap database (www.symmap.org, accessed on 15 October 2024) was employed to obtain further target information corresponding to the detected substances. Ultimately, a total of 253 potential active components and 2021 protein targets were identified across these databases (Appendix A).

### 2.2. The KEGG Enrichment Pathway of the Active Components of Crocus alatavicus

Pathway enrichment analysis was conducted on all identified metabolic targets of *C. alatavicus*. Notably, the top 20 KEGG enrichment pathways associated with disease targets included neuroactive ligand–receptor interactions, prostate cancer, lipid metabolism, and atherosclerosis, as well as calcium signaling pathways, among others, with prostate cancer being the most prominent (Figure 2A). Further pathway enrichment analysis (Figure 2B–E) was performed on active components detected in the corms, roots, leaves, and stems. The results indicated that the relative enrichment of prostate cancer-related pathways in corms was lower compared to that in roots, leaves, and stems. Moreover, the enrichment observed in the roots and stems was the most significant, suggesting that the roots and stems of *C. alatavicus* contained higher concentrations of active ingredients.

### 2.3. Disease Target Prediction and Intersection Analysis

Disease-related targets were predicted using DisGeNET (https://www.disgenet.org/, accessed on 15 October 2024), SymMap (www.symmap.org, accessed on 15 October 2024), and the Therapeutic Target Database (TTD) (https://db.idrblab.net/ttd/, accessed on 15 October 2024). The target information obtained from these databases is presented in Appendix A. To streamline the analysis and minimize potential confounding effects from multiple compounds and disease targets, a more stringent target screening methodology was employed. Specifically, the probability threshold in SwissTargetPrediction was set to be greater than 0.12, resulting in the identification of 1360 targets (Appendix A) from a total of 2021 protein targets.

To establish the relationship between active components and diseases for subsequent network analysis, an intersection analysis of material target information and disease target information was conducted. The results of this intersection analysis following screening are illustrated in Figure 3A. We identified 27 targets associated with prostate cancer from the DisGeNET, SymMap, and TTD databases. A Venn diagram analysis comparing these 27 targets with the 1360 prostate cancer-related targets revealed that 9 targets were shared between the active components of *C. alatavicus* and prostate cancer: *KLK3*, *TP53*, *CCND1*, *EPHB2*, *AR*, *CDH1*, *PIK3CA*, *SRD5A1*, and *CTNNB1*. This finding suggests that nine active components in *C. alatavicus* may potentially interact with targets implicated in prostate cancer pathogenesis. Furthermore, KEGG pathway enrichment analysis performed on these nine common targets demonstrated that the prostate cancer pathway was the most significantly enriched (Figure 3B).

### 2.4. Protein–Protein Interaction (PPI) Analysis

Protein–protein interaction (PPI) analysis was conducted on the 1360 targets obtained previously using the STRING database (http://cn.string-db.org/, accessed on 20 December 2024). Four proteins with the highest number of associations were selected for network mapping: *TP53*, *AKT1*, *STAT3*, and *EGFR* (Figure 3C). Additionally, PPI analysis was performed on the nine shared targets between *C. alatavicus* and prostate cancer (Figure 3D). The results indicated that, with the exception of *SRD5A1*, all other gene targets exhibited potential interactions. The interactions between AR-CTNNB1, CTNNB1-CDH1, and AR-CCND1 ranked among the top three in terms of binding scores. Among the eight gene targets, *PIK3CA*, *TP53*, *CTNNB1*, *CDH1*, and *AR* exhibited high degree values, all equal to 5. These findings suggest that the gene targets of the active components of Crocus sativus are likely to interact effectively with the protein targets associated with prostate cancer.

### 2.5. Construction of Network Diagram

Following the intersection analysis of material targets and disease targets, common targets were selected to construct a network diagram representing the relationships among components, targets, pathways, and diseases. Based on the results shown in Figure 2A and Figure 3B, we focused on prostate cancer as the disease of interest and its associated pathways. There were nine intersecting targets between the active components of *C. alatavicus* and the disease targets related to prostate cancer, with six targets involved in the prostate cancer pathway (Figure 4).

A network diagram of the active components and their corresponding targets was generated using Cytoscape 3.10.3, incorporating 142 chemical components, nine targets, and information on *C. alatavicus*. As illustrated in Figure 4, the network consists of 151 nodes—142 representing active ingredient nodes, 6 representing gene target nodes, 1 for the pathway, 1 for the disease target, and 1 drug node—and 309 edges. All 142 chemical components were associated with the disease and played a role in network construction. In the diagram, hexagonal nodes (outer circle) signify the active ingredients of *C. alatavicus*, while circular nodes (inner circle) denote the common targets.

Through the analysis of network topology parameters, the core targets that play a key regulatory role in the signal pathways related to prostate cancer were identified. These targets are mostly involved in several biological processes closely related to the occurrence and progression of prostate cancer, such as cell proliferation, apoptosis, invasion and metastasis, and the regulation of the tumor microenvironment [18,19]. The results revealed a polypharmacological pattern; single active ingredients exhibited multi-target interactions with multiple proteins, while distinct ingredients shared common protein targets. To systematically characterize these interactions in the context of prostate cancer, we constructed a compound–protein network and analyzed its topological features using the Network Analyzer tool in Cytoscape 3.10.3. Node degree values were calculated to identify hub proteins that may serve as critical therapeutic targets for further investigation. The active components ranked by degree included Capillarisin, Eugenol, 1-(4-Methoxyphenyl)-1-propanol, 2,4,2′,4′-tetrahydroxy-3′-prenylchalcone, and 4-Hydroxymandelonitrile. Notably, among the intersecting targets of the components and diseases, the androgen receptor (AR) displayed the highest degree value, reaching 92.

### 2.6. GO Functional Classification Annotation Analysis

To enhance our understanding of the mechanisms by which *C. alatavicus* may treat prostate cancer, we utilized the DAVID V6.8 to perform Gene Ontology (GO) function and KEGG pathway enrichment analyses on nine core target genes (Figure 5).

The analysis revealed 37 entries related to biological processes (BPs), 19 entries associated with cellular components (CCs), and 8 entries pertaining to molecular functions (MFs) within the GO framework (Figure 5B). Notably, the proportion of biological process entries surpassed the combined total of cellular component and molecular function entries (Figure 5B). Among these results, we highlighted the top six entries in Figure 5C. For instance, the biological processes predominantly included responses to exogenous stimuli and positive regulation of gene expression. These findings effectively demonstrate that *C. alatavicus* possesses potential medicinal value in treating prostate cancer by modulating multiple cellular compartments and related biological processes.

Furthermore, we identified a total of 45 signaling pathways associated with the core targets, with the most prominent pathways presented in Figure 5A. Among the common gene targets linked to both the active components and prostate cancer, the *TP53* and *CCND1* genes were found to participate in the greatest number of disease pathways. Prostate cancer exhibited the highest enrichment of gene targets among the identified diseases. It is noteworthy that, in addition to prostate cancer, the core targets also encompass a variety of other cancers, including endometrial, thyroid, gastric, colorectal, and breast cancers. Collectively, these results indicate that numerous active ingredients in *C. alatavicus* can exert therapeutic effects on prostate cancer through multiple targets and biological functions.

### 2.7. Molecular Docking

Based on the findings from network pharmacology, we screened three primary compounds, Capillarisin, Eugenol, and 4-Hydroxymandelonitrile, and performed molecular docking analyses with their respective proteins: AR (PDB ID: 1E3G), CCND1 (PDB ID: 6P8E), PIK3CA (PDB ID: 8BCY) and KLK3 (PDB ID: 7JOD). The docking scores for each compound–protein interaction are presented in Table 1. Additionally, Figure 6 illustrates the visual representation of the optimal docking configuration between the receptor and ligand.

The results indicate that Capillarisin can interact with the proteins AR, CCND1, and PIK3CA (Figure 6A–C). As illustrated in Figure 6A, Capillarisin forms a single hydrogen bond with each protein of GLU-120, THR-263, and THR-18 in AR, and two hydrogen bonds with LYS-48. Figure 6B demonstrates that Capillarisin establishes one hydrogen bond with each protein of GKU-83 and ASN-66 in CCND1 and two hydrogen bonds with LYS-68. Furthermore, as depicted in Figure 6C, Capillarisin interacts with ARG-969, LEU-972, and LYS-1006 in PIK3CA through a single hydrogen bond.

The results demonstrate that Eugenol can interact with the proteins AR, CCND1, and KLK3 (see Figure 6D–F). As illustrated in Figure 6D, Eugenol forms two hydrogen bonds with THR-110 in AR and one hydrogen bond with ASN-104 and PHE-285. Figure 6E reveals that Eugenol establishes a hydrogen bond with ASN-276 and ASP-234 in CCND1. Additionally, as shown in Figure 6F, the structure of Eugenol interacts with Asn-202 and Ser-135 in KLK3 through a single hydrogen bond.

The results indicate that 4-Hydroxymandelonitrile can interact with PIK3CA (see Figure 6G). As illustrated in Figure 6G, the structure of 4-Hydroxymandelonitrile interacts with PHE-751and LYS-708 through two hydrogen bonds. Lower binding energy values indicate a higher affinity between the receptor and ligand, resulting in a more stable conformation. It is widely accepted that a binding energy of less than −5 kcal/mol signifies strong binding activity between the ligand and receptor [20,21,22,23]. The molecular docking results showed that the binding energy of Capillarisin to AR was less than −5 kcal/mol, indicating that the compound had a high affinity with the protein. Based on this, we predict that Capillarisin may play a key role in the treatment of prostate cancer.

### 2.8. Molecular Dynamics Simulation

#### 2.8.1. Stability Analysis

##### Root Mean Square Deviation (RMSD) Analysis

Root mean square deviation (RMSD) is a commonly used metric for measuring the difference between two structures. It is calculated by comparing the spatial coordinates of corresponding atoms in two molecules, which reflects the conformational similarity between the two molecules. RMSD is used to track the changes in the molecular structure relative to the initial structure during the simulation and to observe whether the amplitude of the changes tends to stabilize. A lower RMSD value indicates that the two structures are more similar. As shown in Figure 7A, the RMSD of the complex structure gradually stabilizes as the simulation progresses, indicating that the structure of the complex gradually becomes stable.

##### Radius of Gyration (Rg) Analysis

The radius of gyration (Rg) is the root mean square distance of all atoms in a molecule relative to the center of mass. It reflects the distribution range of the molecular atoms relative to the center of mass and serves as an important parameter for measuring the overall compactness of the protein–small-molecule structure. A smaller radius of gyration (Rg) indicates that the molecule is more compact, while a larger radius of gyration suggests that the molecular structure is looser and may be in a fluffy state. As can be seen from Figure 7B, the Rg of the complex gradually stabilizes as the simulation progresses, which indicates that the structure of the complex is gradually stabilizing.

##### Analysis of the Evolution of the Center of Mass

To analyze the state of the small molecule on the surface of the protein, the initial docking site of the small molecule was obtained, and the distance between the center of mass of the residues at the initial docking site and the center of mass of the small molecule was analyzed. Additionally, the distance between the center of mass of the small molecule and that of the protein was also analyzed. Through the analysis of these distances, the binding state between the small molecule and the protein could be determined. As shown in Figure 7C, the distance between the small molecule and the center of the protein, as well as the distance between the small molecule and the binding site, gradually stabilized. This indicates that the small molecule stably binds to the binding site of the protein, and the binding between the small molecule and the protein gradually becomes stable.

##### Analysis of Buried Solvent Accessible Surface Area (Buried SASA)

The Buried Solvent Accessible Surface Area (Buried SASA) is a measure used to evaluate the regions within a molecule or molecular complex that are buried inside and not directly accessible to the solvent. A larger value of Buried SASA indicates stronger intermolecular interactions and a larger contact area. As shown in Figure 7D, the Buried SASA gradually stabilizes, suggesting that the contact area between the small molecule and the protein is gradually stabilizing, and their binding is becoming increasingly stable.

##### Binding Conformation Superposition

The simulation trajectory was processed, and the simulated conformations were superimposed. The results are shown in Figure 7E. The small molecule shows a high degree of superposition, which indicates that the small molecule remains bound to the protein throughout.

#### 2.8.2. Analysis of the Hydrogen Bond Interaction Between the Small Molecule and the Protein

##### Evolution of the Number of Hydrogen Bonds

Hydrogen bond interaction is an important force in the binding between a protein and a ligand. Hydrogen bonds are related to electrostatic interactions and can reflect the strength of electrostatic interactions. As shown in Figure 8C, the number of hydrogen bonds between the small molecule and the protein basically fluctuates between 0 and 4.

#### 2.8.3. Analysis of the Interaction Between the Small Molecule and the Protein upon Binding

##### Analysis of Electrostatic and Van Der Waals Interactions

Without considering the solvation effect, the van der Waals force and electrostatic interaction force between the small molecule and the protein in the complex were calculated, and the changes in the binding force during the simulation process were analyzed. Among them, VDW represents the van der Waals force and hydrophobic interaction, ELE represents the electrostatic interaction, and binding is the sum of VDW and ELE, which can represent the binding energy between the small molecule and the protein without considering the solvation effect. As shown in Figure 8A, VDW and ELE in the complex basically remain stable as the simulation progresses, which indicates that the binding between the small molecule and the protein basically remains stable.

##### Binding Energy Analysis

When considering the solvation energy, taking RMSD, Rg, distance, Buried SASA and interaction energy into comprehensive consideration, the complex trajectory in the stable state was selected and calculated using the Molecular Mechanics/Poisson–Boltzmann Surface Area (MM-PBSA) method to obtain the energy terms related to the binding energy, as shown in Table 2. Among them, ΔEele represents the electrostatic interaction between the small molecule and the protein, ΔEvdw represents the van der Waals interaction, ΔEpol represents the polar solvation energy, which can represent the electrostatic potential energy, ΔEnonpol represents the non-polar solvation energy, which can represent the hydrophobic interaction, ΔEMMPBSA is the sum of ΔEele, ΔEvdw, ΔEpol and ΔEnonpol, and the Gibbs Binding Energy ΔGbind is the sum of ΔEMMPBSA and −TΔS. (Note: Since the calculation of −TΔS has a large error, in the comparison of binding energy, this term is often not counted, and ΔEMMPBSA can be directly used as the binding energy; in addition, there is also a large error in the calculation of the polar solvation energy ΔEpol, and more attention can be paid to the other energy terms such as ΔEele, ΔEvdw and ΔEnonpol). Through the analysis of Table 2, the van der Waals interaction energy ΔEvdw in the complex is higher than the electrostatic interaction ΔEele, and both of them are higher than the hydrophobic interaction ΔEnonpol. Therefore, in the composition of the binding energy, the van der Waals interaction plays a major role, the electrostatic interaction plays a secondary role, and the hydrophobic interaction plays a supplementary role. The ΔEMMPBSA between the small molecule and the protein is −37.394 ± 2.244 kJ/mol, indicating that the binding energy and affinity between the two are relatively high.

##### Residue Contribution Analysis

The binding energy ΔEMMPBSA is decomposed to obtain the contribution of each amino acid to the overall binding energy, and the important amino acids in the protein are evaluated. The residues in the protein that contribute significantly to their respective binding energies are shown in Figure 8B. It can be seen from this figure that the key amino acids in the protein that bind to the small molecule include PRO-801, TRP-751, etc.

##### Structural Analysis

The conformation when the simulation is stable is selected, and its structure and interactions are analyzed. As shown in Figure 8D, the amino acid TRP-751 in the protein forms a hydrogen bond with the small molecule, PRO-801 forms a Pi-Alkyl hydrophobic interaction with the small molecule, and amino acids such as THR-800 and PHE-804 form van der Waals interactions with the small molecule.

### 2.9. KEGG Annotation Pathway Analysis

In the context of the prostate cancer pathway (Figure 9), we identified that six of the intersecting genes were enriched in the PI3K-Akt signaling pathway, MAPK signaling pathway, and P53 signaling pathway. These genes include *CTNNB1*, *TP53*, *KLK3*, *PIK3CA*, *CCND1*, and *AR*. Notably, *CTNNB1* encodes Catenin Beta 1, which plays a role in the PI3K-Akt signaling pathway. The *PIK3CA* gene is recognized as an oncogene primarily involved in the regulation of cell proliferation, differentiation, and apoptosis, with its encoded protein being a component of the *PI3K* enzyme complex. Additionally, the protein encoded by *CCND1* is Cyclin D1, which is crucial for cell cycle regulation. Prostate cancer is particularly dependent on the androgen receptor (AR), a transcription factor that regulates several biological pathways essential for the growth and survival of prostate cancer cells. Consequently, it can be inferred that numerous active ingredients found in *C. alatavicus* may exert therapeutic effects on prostate cancer by modulating protein expression associated with these pathways.

## 3. Discussion

Utilizing a network pharmacology approach, we identified 142 active components of *C. alatavicus*, among which five key constituents were highlighted: Capillarisin, Eugenol, 1-(4-Methoxyphenyl)-1-propanol, 2,4,2′,4′-tetrahydroxy-3′-prenylchalcone, and 4-Hydroxymandelonitrile. Notably, the efficacy of Eugenol in the treatment of prostate cancer has been validated through clinical studies [24]. We identified nine significant targets relevant to the treatment of prostate cancer, including *KLK3*, *CCND1*, *EPHB2*, and *SRD5A1*, as well as five core targets: *PIK3CA*, *TP53*, *CTNNB1*, *CDH1*, and *AR*. Our analysis revealed 37 associated biological processes, such as responses to exogenous stimuli and positive regulation of gene expression, along with 20 important signaling pathways, including neuroactive ligand–receptor interactions, prostate cancer pathways, lipid metabolism and atherosclerosis, and calcium signaling pathways. Through molecular docking, we further confirmed that Capillarisin exhibits a strong binding affinity for androgen receptors (ARs); further molecular dynamics simulations and binding free energy analyses showed that the binding between ARs and Capillarisin is stable, and the binding interaction is strong, indicating that Capillarisin is a significant potential candidate for the treatment of prostate cancer.

Research has demonstrated that prostate cancer (PCa) is characterized by genomic alterations that play a pivotal role in carcinogenesis. In particular, the dysregulation of genes associated with the androgen receptor (AR) and AR signaling pathways is central to the pathogenesis of PCa. Additionally, alterations in the PI3K/Akt pathway represent the second most frequently observed aberration in metastatic prostate cancer [25,26,27,28,29]. The most prevalent genetic abnormalities identified in prostate cancer include point mutations in genes such as *SPOP*, *FOXA1*, and *TP53*, alongside copy number alterations (CNAs) affecting *AR*, *MYC*, *RB1*, *PTEN*, *CHD1*, and fusion genes associated with the ETS (E26 transformation-specific) family [26,30,31,32,33,34,35,36]. It is noteworthy that the alterations observed in the components of the androgen receptor (AR) and PI3K/Akt pathways appear to be a consequence of ongoing systemic therapy [37,38]. A total of 91 active components, including 1-(4-Methoxyphenyl)-1-propanol, 18-Demethylparaensidimerin C, and 1-Decanol, have been identified as capable of binding to androgen receptor (AR) targets. These components promote cell proliferation and survival through various pathways, including the PI3K/Akt pathway and MAPK signaling pathway.

Through our analysis of the KEGG pathways, we found that the active ingredients of *C. alatavicus* primarily exert their therapeutic effects on prostate cancer through mechanisms involving neuroactive ligand–receptor interactions, lipid metabolism, and atherosclerosis, as well as calcium signaling pathways. All identified targets interact with the neuroactive ligand–receptor interaction pathway, including the androgen receptor (*AR*), *CDH1*, *CTNNB1*, *TP53*, and *PIK3CA*. Consequently, the active components of *C. alatavicus* may inhibit the proliferation of cancer cells by modulating the activity of these receptors, thereby achieving a therapeutic effect. Furthermore, the AR may interact with the receptors of neuroactive ligands to form a regulatory network. For instance, in prostate cancer, the interaction between ARs and other hormone receptors, such as estrogen receptors, can influence tumor growth and progression [39,40]. *CDH1* plays a crucial role in the signal transduction pathways associated with cancer, particularly in tumors linked to neuroactive ligand–receptor interactions. The expression level of *CDH1* can significantly influence tumor progression and metastasis, as it is associated with the abnormal activation of neuroactive ligand–receptor signaling pathways [41].

The polymorphism of *CTNNB1* is associated with cancer risk, suggesting that it may affect the occurrence and development of tumors by regulating the role of neuroactive ligands in the tumor microenvironment [42]. In the case of severe DNA damage, *TP53* can activate the apoptotic process and promote cell self-extinction to prevent tumor formation. *TP53* protein can regulate a variety of genes related to DNA repair, ensuring that cells can repair their genomes in a timely manner after damage. *TP53* mutant cells may change the tumor microenvironment and affect the interaction between nerve cells and tumor cells. This interaction may affect tumor growth and metastasis through the neuroactive ligand–receptor pathway. Similarly, there is a cross between the activation of *PIK3CA* and the neuroactive ligand–receptor interaction pathway. For example, the activation of the PI3K/Akt pathway can enhance the response of cells to neurotransmitters, regulate cell proliferation and survival, and thus affect the role of neuroactive substances in the tumor microenvironment [43].

The core value of network pharmacology is that TCM efficacy often stems from multi-component, multi-target synergies. Our “active ingredient-target-disease-pathway” network shows *Crocus alatavicus* components (e.g., Capillarisin, Eugenol) link multiple targets, with complex target–target interactions. In prostate cancer pathways (PI3K-Akt, MAPK), key targets like AR, PIK3CA, and TP53 are co-regulated by multiple components, indicating synergies via shared targets or pathways. Structural analysis reveals complementary binding; Capillarisin forms AR hydrogen bonds, while Eugenol interacts via hydrophobic forces. These enhance ligand–receptor stability and AR pathway regulation, embodying synergistic mechanisms. Future studies using cell and in vivo models can explore component synergies, deepening the understanding of *C. alatavicus*’s medicinal potential and strengthening theoretical support for drug development.

This study used computational methods to show potential interactions between *Crocus alatavicus* components (like Capillarisin) and prostate cancer targets (such as ARs). Yet empirical validation is vital to link in silico predictions with biological facts. Future research could start by isolating and confirming key constituents like Capillarisin and Eugenol from plant extracts via HPLC and NMR, matching computational models. In vitro experiments on prostate cancer cell lines (e.g., LNCaP, PC-3) will verify the regulatory effects on AR signaling pathways, assessing PSA expression, cell processes, and using siRNA to distinguish interactions. In vivo evaluation in nude mouse xenograft models will assess antitumor activity and safety, combined with LC-MS/MS for pharmacokinetic profiling. Also, studies on synergistic effects of multi-constituent combinations on ARs, PI3K-Akt, etc., quantified by isobolographic analysis, and optimization of Capillarisin analogs’ structure–activity relationships will be performed to address pharmacokinetic issues and improve druggability. These experiments will help shift *C. alatavicus* from “computational prediction” to “experimental validation”, aiding its development as an AR-targeted natural probe and promoting traditional medicinal plant research.

## 4. Materials and Methods

### 4.1. Screening for Active Ingredients and Targets Prediction

The data of 1900 metabolites of *C. alatavicus* required for this study were sourced from the previous widely targeted metabolomics study conducted by our research team. Data collection was performed using Ultra Performance Liquid Chromatography (UPLC) (ExionLC™ AD, https://sciex.com.cn/, accessed on 10 August 2024) and tandem mass spectrometry (MS/MS). Substances were qualitatively identified according to the secondary spectrum information based on the self-established database MWDB (metware database). The mass spectrometry data were processed using the software Analyst 1.6.3. (The samples of *C. alatavicus* were collected from Yili Botanical Garden in Xinjiang, Northwest China. In mid-April 2023, several whole plants with intact root soil were transferred to the laboratory. One gram of roots, stems, leaves, and corms was separately collected, washed multiple times with deionized distilled water, wrapped in aluminum foil, and immediately stored in a −80 °C ultra-low temperature freezer.)

Active compounds of *C. alatavicus* were systematically identified and computationally analyzed using TCMSP, SwissDrugDesign, and SymMap platforms. Initial screening in TCMSP (https://old.tcmsp-e.com/tcmsp.ph, accessed on 15 October 2024) [44] applied pharmacokinetic criteria (OB ≥ 30%, DL ≥ 0.18) to filter bioactive components, followed by UniProt-based target standardization. These compounds underwent comprehensive drug-likeness evaluation via SWISSADME (http://www.swissadme.ch/, accessed on 15 October 2024), assessing gastrointestinal absorption and compliance with Lipinski, Ghose, Veber, Egan, and Muegge rules [45]. Compounds meeting both high gastrointestinal absorption and ≥2 drug-likeness criteria were then submitted to SwissTargetPrediction (http://www.swisstargetprediction.ch/, accessed on 15 October 2024) for target prediction and validated against UniProt annotations. Final integration of compound–target networks and pathway analysis was performed using SymMap (www.symmap.org), leveraging its multi-source pharmacological database [46].

### 4.2. Identification of Disease Targets

Disease-specific targets relevant to the therapeutic effects of traditional Chinese medicine (TCM) were systematically retrieved from three integrated databases: DisGeNET, SymMap, and TTD. The DisGeNET platform (https://www.disgenet.org/, accessed on 15 October 2024) provided gene–disease associations filtered by confidence scores ≥ 0.5 and experimental evidence (GWAS, expression QTLs). SymMap (www.symmap.org, accessed on 15 October 2024) was used to map these genes onto pathological pathways (KEGG, Reactome) through network analysis, while TTD (https://db.idrblab.net/ttd/, accessed on 15 October 2024) contributed validated drug targets linked to clinical outcomes [47,48,49].

### 4.3. Construction of Active Ingredient–Target–Pathway–Disease Network

Bioinformatics Cloud Platform (http://www.bioinformatics.com.cn/, accessed on 15 October 2024) was utilized to identify the intersection of potential drug targets and disease-related targets. The active ingredients of the drug, along with the identified cross-targets, were uploaded to Cytoscape 3.10.3 software to generate a network diagram illustrating the relationships among active ingredients, targets, pathways, and diseases. Key chemical components and targets were subsequently identified through network topology analysis.

### 4.4. Target Screening and Construction of a Protein–Protein Interaction Network

The cross-target protein list was submitted to the STRING database (http://cn.string-db.org/, accessed on 20 October 2024) to construct a protein–protein interaction (PPI) network. The topological analysis of the PPI network was conducted using Cytoscape v3.10.3 software. In this network, nodes were defined as proteins, and edges were defined as the interactions between proteins. The degree value of each node was calculated via the built-in network analysis function of this software. Here, network topology refers to the geometric arrangement of nodes (proteins) and edges (interactions) in the PPI network. In this study, we focused on the topological parameter of node degree, which represents the number of interactions of each protein with other proteins.

### 4.5. Functional Enrichment Analysis and Pathway Enrichment Analysis of Gene Ontology Based on Kyoto Encyclopedia of Genes and Genomes

Gene Ontology (GO) functional enrichment analysis and Kyoto Encyclopedia of Genes and Genomes (KEGG) pathway enrichment analysis were conducted on the cross-targets using the MetScape database. This analysis aimed to identify the primary action pathways associated with the active components of *C. alatavicus* in the treatment of prostate cancer.

### 4.6. Molecular Docking

Molecular docking methods were utilized to investigate whether the core components of *Crocus alatavicus*, identified through network pharmacology, interact with key proteins. The procedure was conducted as follows: based on the results of the previous network pharmacology screening, the three-dimensional structure of the target protein was retrieved from the Protein Data Bank (PDB) (http://www.rcsb.org/, accessed on 2 December 2024) and the corresponding PDB format file was downloaded [50]. The protein was then configured in AutoDock4 [51] as follows: remove the water molecules, replace them with hydrogen atoms, designate the protein as a receptor, and save the structure as a PDBQT protein receptor file. The structures of the drug molecules were downloaded from the TCMSP and PubChem databases (https://pubchem.ncbi.nlm.nih.gov/, accessed on 15 October 2024) [52]. Similarly, in AutoDock4, the drug was prepared using the following steps: water molecules were removed, hydrogen atoms were added, and the drug was designated as a ligand. The ligand structures were converted to the PDBQT format using Open Babel to be compatible with the AutoDock 1.5.6 software. Subsequently, AutoDock was employed for molecular docking to calculate the minimum binding energy and identify the interactions between ligands and proteins. The target protein served as the center of the grid, with the center coordinates (center x/y/z) and box size parameters (size x/y/z) adjusted to ensure that the entire protein was encompassed by the docking box [53]. Molecular docking was conducted in AutoDock4 by identifying protein macromolecules and introducing small drug molecules, as well as configuring the operational methods and docking parameters. The PDBQT format was utilized to calculate the minimum binding energy. OpenBabelGUI 3.1.1 software [54] was used to convert the composite from PDBQT format to PDB format. Finally, for visualization purposes, the composite PDB format file was imported into PyMOL 4.6 [55].

### 4.7. Molecular Dynamics Simulation

Molecular dynamics (MD) simulation was carried out using the Gromacs 2022 program. The General Amber Force Field (GAFF) was employed for small molecules, while the AMBER14SB force field and the TIP3P water model were used for proteins. The files of the protein and the small-molecule ligand were merged to construct the simulation system of the complex. The simulation was conducted under constant temperature and pressure conditions, as well as periodic boundary conditions. During the MD simulation, the LINCS algorithm was used to constrain all hydrogen bonds involved, and the integration time step was set to 2 femtoseconds (fs). The Particle-mesh Ewald (PME) method was applied to calculate the electrostatic interactions, with a cutoff value set at 1.2 nanometers (nm). The cutoff value for non-bonded interactions was set at 10 angstroms (Å), and it was updated every 10 steps. The V-rescale temperature coupling method was adopted to control the simulation temperature at 298 Kelvin (K), and the Berendsen method was used to control the pressure at 1 bar. At 298 K, a 100 picosecond (ps) NVT (constant number of particles, volume, and temperature) and NPT (constant number of particles, pressure, and temperature) equilibration simulation was performed, and a 100 nanosecond (ns) MD simulation was carried out for the complex system. The conformation was saved every 10 ps. After the simulation was completed, visual molecular dynamics (VMD) and PyMOL were used to analyze the simulation trajectory, and the g_mmpbsa program was utilized to conduct a Molecular Mechanics/Poisson–Boltzmann Surface Area (MMPBSA) binding free energy analysis between the protein and the small-molecule ligand.

## 5. Conclusions

Using network pharmacology, this study revealed that five core bioactive components of *C. alatavicus* (e.g., Capillarisin) target androgen receptor (AR)/PI3K-Akt pathways in prostate cancer. Molecular modeling validated stable AR binding of Capillarisin (supported by ΔE_MMPBSA_ analysis), with multi-component synergy identified across 20 signaling axes (notably neuroactive ligand-receptor interactions) modulating tumor microenvironments. These findings provide computational evidence for *C. alatavicus* as a potential AR-targeted natural agent. Future validation requires HPLC-NMR verification of key constituents, in vitro/in vivo confirmation of AR pathway regulation (via LNCaP/PC-3 cell models and xenograft assays), and structure-activity optimization to advance translational potential from in silico prediction to experimental translation.

## Figures and Tables

**Figure 1 ijms-26-03533-f001:**
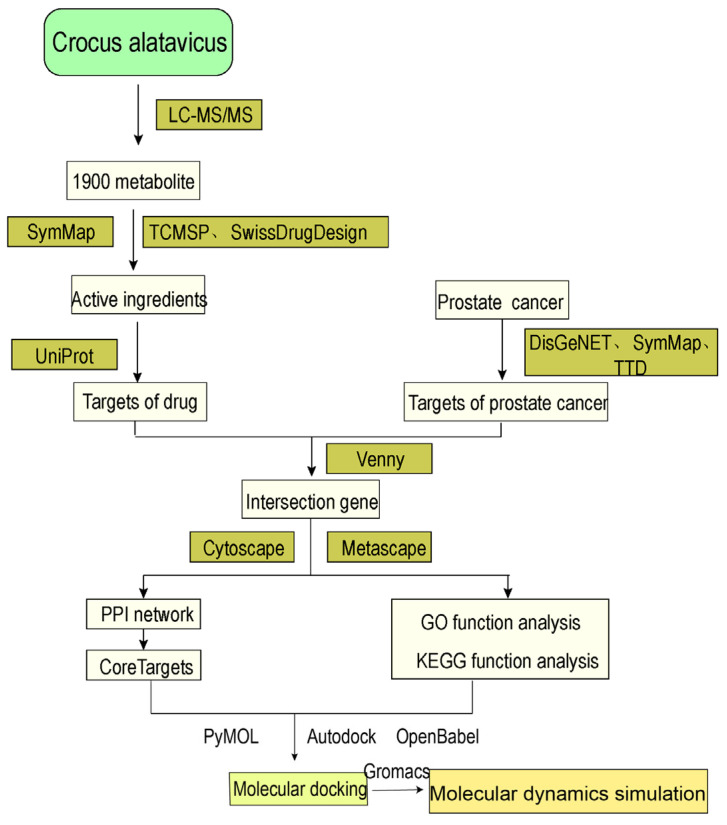
A flowchart illustrating the investigating trajectory from the original metabo lome to the discovery of core disease related target metabolites.

**Figure 2 ijms-26-03533-f002:**
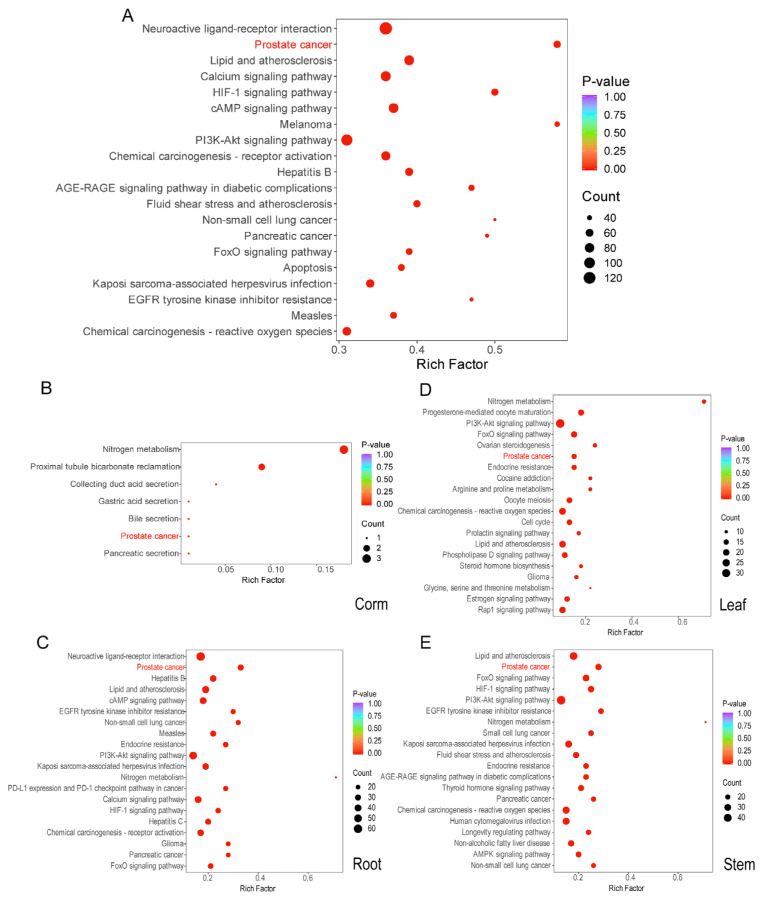
Enrichment map of the common target pathways of medicinal ingredients in *C. alatavicus* and prostate cancer. (**A**) KEGG pathway enrichment analysis of all active ingredients of *C. alatavicus*; (**B**–**E**) represent the KEGG pathway enrichment analysis of the active ingredients in the corm, root, leaf, and stem, respectively. The abscissa represents the Rich Factor corresponding to each pathway, and the ordinate represents the pathway name (sorted according to the *p*-value). The color of the dots reflects the magnitude of the *p*-value. The redder the color, the more significant the enrichment. The size of the dots indicates the number of genes annotated on the pathway.

**Figure 3 ijms-26-03533-f003:**
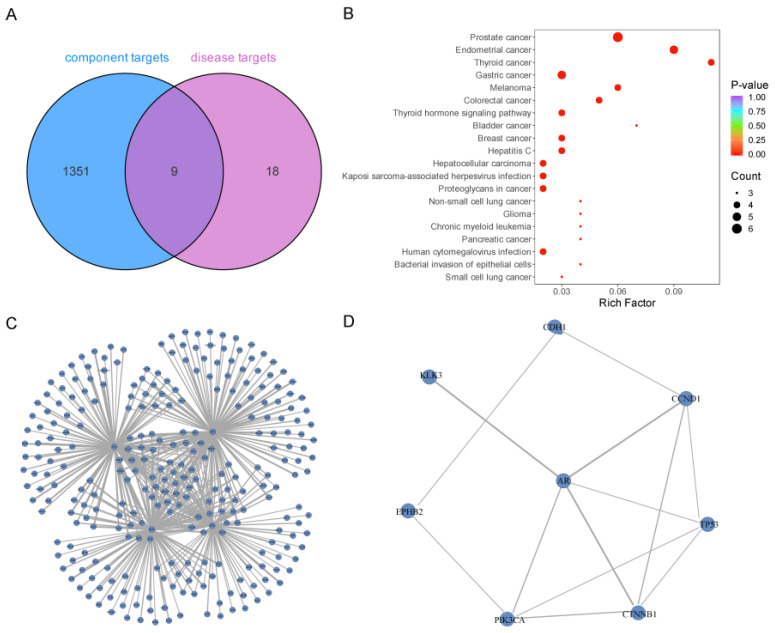
Enrichment analysis and protein interaction of common target pathways between active components and prostate cancer. (**A**) represents the Venn diagram analysis of the active ingredients of *Crocus alatavicus* and prostate cancer. (**B**) shows the enrichment pathway analysis of the common targets between the active ingredients of *C. alatavicus* and prostate cancer (sorted according to the *p*-value). The color of the dots reflects the magnitude of the *p*-value, with redder dots indicating more significant enrichment. The size of the dots represents the number of genes annotated on the corresponding pathway. In (**C**), a protein–protein interaction (PPI) analysis was performed on 1360 targets corresponding to the active ingredients of *C. alatavicus*. Four proteins with the highest number of associations were selected for network mapping. Each node in the figure represents a protein, and the lines connecting the nodes indicate the existence of interactions between proteins. The thickness of the lines represents the strength of the interactions, with thicker lines indicating stronger binding. (**D**) is the PPI analysis diagram of the nine common targets between *C. alatavicus* and prostate cancer. The nodes represent the proteins corresponding to the gene targets, and the edges represent the interactions between proteins. The thickness of the edges is differentiated according to the binding scores of the interactions.

**Figure 4 ijms-26-03533-f004:**
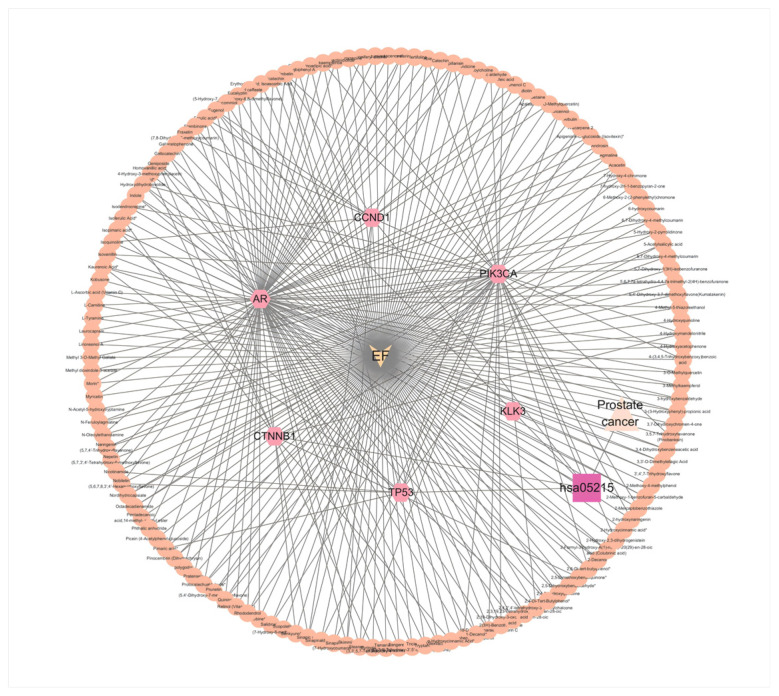
Component–target–pathway–disease network diagram. The circle in the outer ring represents the active ingredients of *C. alatavicus*. The circle in the inner ring represents the common targets shared by the diseases involved in the prostate cancer pathway and the active ingredients. The inverted triangle at the exact center represents the drug node. The triangle represents prostate cancer, and the square represents the pathway.

**Figure 5 ijms-26-03533-f005:**
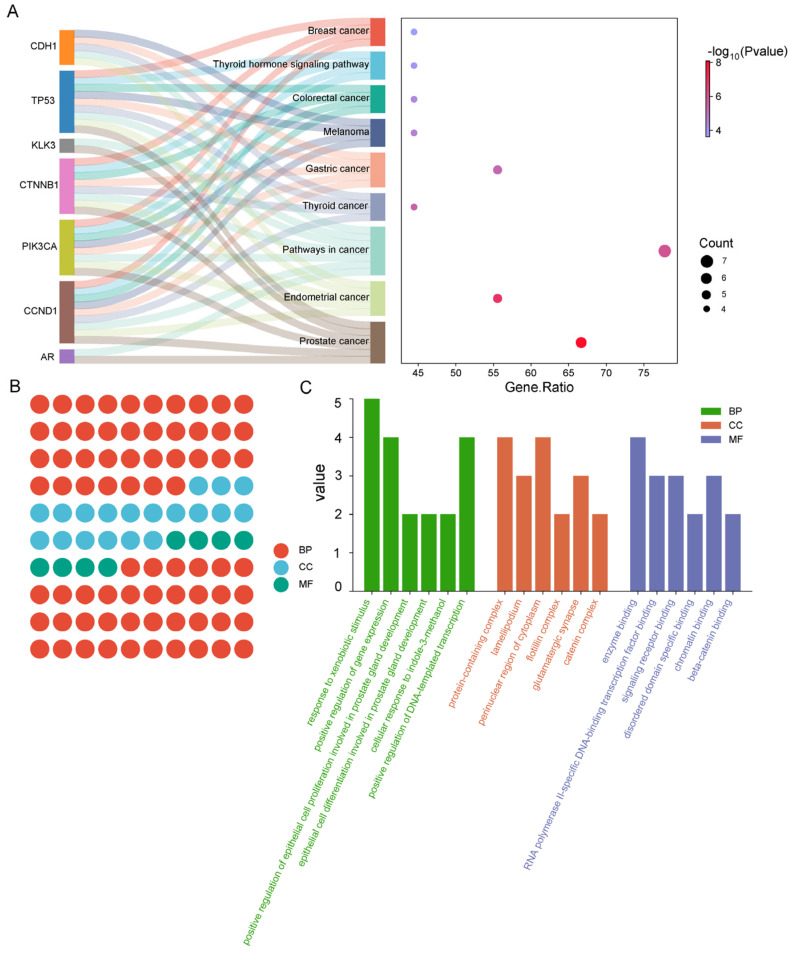
GO functional classification annotation analysis. (**A**) The leftmost part represents the common gene targets related to both the active ingredients of *C. alatavicus* Regel et Sem. and prostate cancer. The right side represents the enrichment of signaling pathways associated with the core targets. The size of the circles indicates the number of genes enriched in the pathways, and the redder the color, the more significant the enrichment. (**B**) Gene Ontology (GO) functional analysis of the nine core target genes. Red represents biological processes, blue represents cellular components, and green represents the entries related to molecular functions (MF). (**C**) Kyoto Encyclopedia of Genes and Genomes (KEGG) pathway enrichment analysis: the horizontal axis. Green represents the entries related to biological processes (BPs), orange represents the entries related to cellular components (CCs), and purple represents the entries related to molecular functions (MFs). The vertical axis represents the number of genes being enriched.

**Figure 6 ijms-26-03533-f006:**
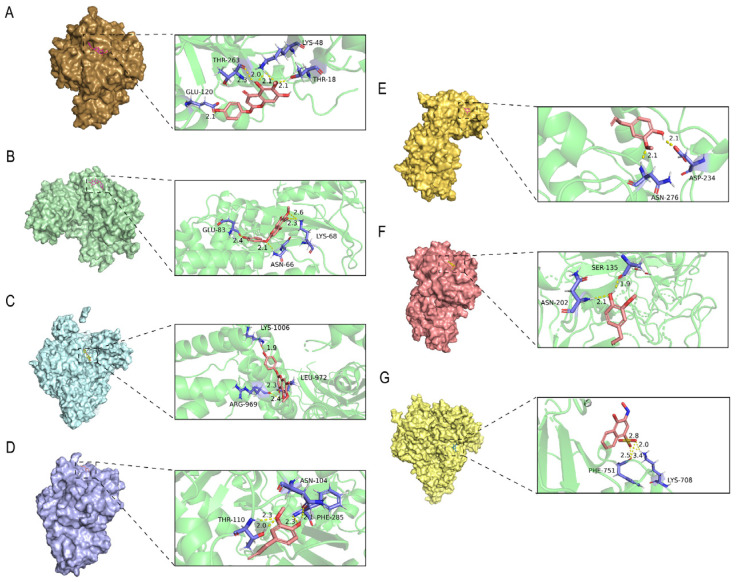
Molecular docking results of main chemical components of *C. alatavicus*. (**A**) Capillarisin-AR(1E3G); (**B**) Capillarisin-CCND1(6P8E); (**C**) Capillarisin-PIK3CA(8BCY); (**D**) Eugenol-AR(1E3G); (**E**) Eugenol-CCND1(6P8E); (**F**) Eugenol-KLK3(7JOD); (**G**) 4-Hydroxymandelonitrile- PIK3CA(8BCY). On the left is a visualization of the small molecule within the protein pocket, and on the right is the docking of the protein with the small molecule. The small molecule compound is colored red, and the amino acid groups of the protein are colored purple, interacting via hydrogen bonds. specific grid parameters in molecular docking: AR: X = 5.756, Y = 60.43, Z = 10.928 (grid size: 21 × 21 × 21 Å); PIK3CA: X = 16, Y = 14, Z = 13 (grid size: 30 × 30 × 30 Å); CCND1: X = 24, Y = 13, Z = 32 (grid size: 30 × 30 × 30 Å); KLK3 = X = 12.3, Y = 34.7, Z = 56.2 (grid size: 22.5 × 22.5 × 22.5 Å).

**Figure 7 ijms-26-03533-f007:**
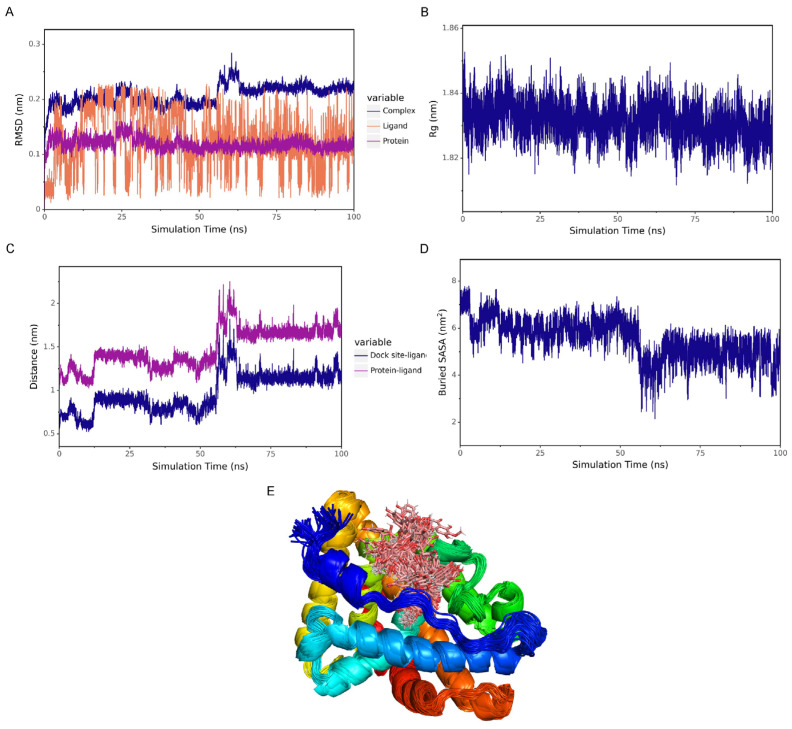
Analysis of the molecular dynamics characteristics of the interaction between Capillarisin and AR. (**A**) The root mean square deviation (RMSD) of the Capillarisin–AR complex, the protein (AR) and the small-molecule ligand (Capillarisin); (**B**) the radius of gyration (Rg) of the Capillarisin–AR complex; (**C**) the distance between the binding site of the AR protein and the small molecule Capillarisin (Dock site–ligand); (**D**) the Buried Solvent Accessible Surface Area (Buried SASA) between the small molecule Capillarisin and the AR protein; (**E**) superposition of the simulated conformations.

**Figure 8 ijms-26-03533-f008:**
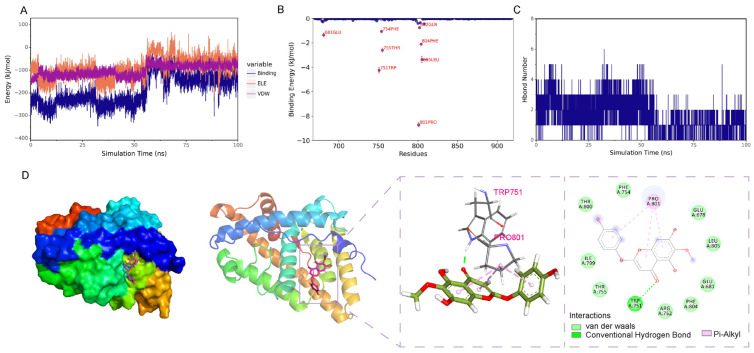
Analysis of the quantitative and dynamic characteristics of the binding between Capillarisin and AR. (**A**) The binding energies of Van der Waals (VDW) and Electrostatic (ELE) between the small molecule Capillarisin and the AR protein. The horizontal axis represents the simulation duration, and the vertical axis represents the energy. (**B**) The contribution of amino acids to the binding energy. The horizontal axis represents the amino acid residues, and the vertical axis represents the binding energy. (**C**) The variation in the number of hydrogen bonds (Hbond number). The horizontal axis represents the simulation duration, and the vertical axis represents the number of hydrogen bonds. (**D**) The interaction between the AR protein and the small molecule Capillarisin.

**Figure 9 ijms-26-03533-f009:**
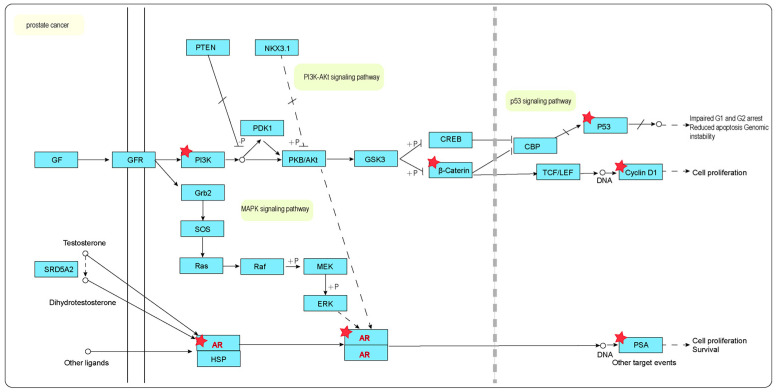
Pathway enrichment analysis and protein–protein interaction among active components and common targets in prostate cancer. The solid arrow represents a relatively clear functional pathway, while the dashed arrow indicates an indirect functional relationship, a potential connection, or a functional pathway that has not been fully elucidated yet. The circle represents a small-molecular compound, and the end of the straight line or the dashed line implies the blocking of this reaction. The proteins corresponding to the red five-pointed stars are those encoded by the genes PIK3CA, CTNNB1, TP53, CCND1, AR, and KLK3 (the intersection target genes between the active ingredients of *C. alatavicus* and prostate cancer diseases). PIK3CA is one of the genes encoding the PI3K protein. CTNNB1 encodes the β-catenin protein. TP53 encodes the P53 protein. CCND1 encodes the Cyclin protein. The AR gene encodes the AR protein. KLK3 encodes PSA (prostate-specific antigen).

**Table 1 ijms-26-03533-t001:** The binding energy of compound and core targets (kcal/mol).

Target	PDB ID	Target Structure	Ligand	Affinity(kcal/mol)
AR	1E3G	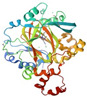	Capillarisin	−6.76
Eugenol	−5.43
CCND1	6P8E	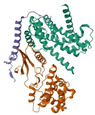	Eugenol	−4.05
Capillarisin	−4.52
PIK3CA	8BCY	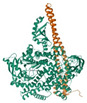	4-Hydroxymandelonitrile	−5.05
Capillarisin	−3.96
KLK3	7JOD	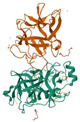	Eugenol	−5.37

**Table 2 ijms-26-03533-t002:** Binding energy and its composition under the stable state (unit: kJ/mol).

Complex	ΔE_vdw_	ΔE_ele_	ΔE_pol_	ΔE_nonpol_	ΔE_MMPBSA_	−TΔS	ΔG_bind_ *
AR–Capillarisin	−85.595 ± 1.633	−25.637 ± 2.759	86.711 ± 4.505	−12.932 ± 0.28	−37.394 ± 2.244	22.579 ± 1.046	−14.814 ± 3.282

* ΔG_bind_ = ΔE_vdw_ + ΔE_ele_ + ΔE_pol_ + ΔE_nonpol_ − TΔS.

## Data Availability

All the data generated or analyzed during the current study are included in the manuscript.

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
