# Peer review of "Network Pharmacology and Molecular Dynamics Identified Potential Androgen Receptor-Targeted Metabolites in Crocus alatavicus"

_ijms, 2025, doi:10.3390/ijms26083533_

Round 1
Reviewer 1 Report
Comments and Suggestions for Authors
1. In the Introduction section, the two paragraphs (lines 51–71) are not appropriately structured as an introduction. They read more like a detailed methodology of the tools used in this study. The authors should revise them to align with the purpose of an introduction.
2. The authors should review the sentence in line 93: 'In this study, 1,900 metabolites derived from the extensive targeted…' and again in line 97: 'In this study, 1,900 metabolites of C. alatavicus…' as they are not technically consistent. Please clarify and ensure accuracy in describing the metabolites.
3.The sentence 'Utilizing LC-MS/MS, a total of 1,900 metabolites from white crocus were identified' is misleading, as this study is entirely computational.
4. Revise the Methods section titled '2.2. Targets Associated with Active Ingredients' for better clarity and organization, or consider merging it with another relevant section to enhance the flow of the manuscript.
5. The sentence in line 254, 'Network topology parameters were analyzed for the treatment of prostate cancer,' is not scientifically accurate.
6.In lines 137–138, the sentence 'Subsequently, we analyzed the topology of the PPI network and calculated the topological parameter values' lacks clarity. The authors should specify the method used to analyze the topology and elaborate on the meaning of 'topology' in this context to enhance scientific accuracy.
7. The Molecular Docking section should be revised for clarity and accuracy. The authors should provide the PDB IDs of the target proteins. The sentence 'The PDBQT format was utilized to calculate the minimum binding energy' is incorrect, as PDBQT is only a compatible format for docking in AutoDock. Additionally, the sentence describing the docking grid setup should be revised to include specific center coordinates (X, Y, Z) and grid box dimensions (X, Y, Z) to ensure clarity and reproducibility.
8.The authors should provide high-resolution versions of Figures 3 and 5 with clear labels and descriptions.
9. In Table 2, the reported binding energies, such as -2.24 kcal/mol, are not reliable, as AutoDock typically yields binding energies in the range of -6 to -10 kcal/mol for strong interactions. Binding affinities below -4 kcal/mol generally indicate weak or non-specific interactions, raising concerns about the validity of these docking results. The authors should re-evaluate the docking protocol, ensure proper grid box settings, exhaustiveness, and validation of the docking parameters to obtain more reliable and meaningful binding affinities.
10. For better authenticity and robustness of the binding interactions, as well as their validation, the authors should perform molecular dynamics (MD) simulations and binding free energy calculations to further confirm the stability and reliability of the docking results.
Author Response
Comment 1: In the Introduction section, the two paragraphs (lines 51–71) are not appropriately structured as an introduction. They read more like a detailed methodology of the tools used in this study. The authors should revise them to align with the purpose of an introduction.
We sincerely appreciate this comment. The introductory section of the manuscript has been systematically revised to enhance logical coherence and expand the scientific context. including the deletion of the pharmacy database section, the incorporation of a comprehensive discussion on the pharmacological characteristics of natural products in drug discovery, and the restructuring of paragraph sequences to enhance logical coherence. Specifically, here we briefly summarized each of the introductory sections to make it clear:
Paragraph 1: Integrated a brief review of natural product drug discovery ï¼›
Paragraph 2: Strengthened the taxonomic and ethnopharmacological background of Crocus alatavicus;
Paragraph 3: Reconstructed the rationale for applying network pharmacology and computational tools of structural biology, positioning this approach as a bridge between traditional herbal medicine and modern molecular/structural biology.
Paragraph 4: Elaborated on the pathological basis and therapeutic challenges of prostate cancer, highlighting the limitations of conventional treatmentsï¼›
Paragraph 5: Scientific issues and brief objectives of the study.
Comment 2: The authors should review the sentence in line 93: 'In this study, 1,900 metabolites derived from the extensive targeted…' and again in line 97: 'In this study, 1,900 metabolites of C. alatavicus…' as they are not technically consistent. Please clarify and ensure accuracy in describing the metabolites.
Thank you for the valuable comments. You pointed out the issue of inconsistent technical descriptions regarding metabolites in lines 93 and 97, which is of great significance for improving the quality of the paper.
Here, we would like to explain to you that the data from the widely targeted metabolomics study is derived from another metabolomics research paper of ours focusing on its metabolomic changes following cold treatment. And this paper on network pharmacology conducts research from a different point of view based on these 1,900 metabolites. To avoid confusing the readers, we have made the following revisions to the paper:
Unify the expression: We have unified the statements in line 93, "A total of 1,900 metabolites were identified in this study through widely targeted metabolomics," and line 97, "1,900 metabolites of Crocus alatavicus were analyzed in this study," and revised them to "Based on the widely targeted metabolomics study where 1,900 metabolites of Crocus alatavicus were identified, network pharmacology research was carried out." (line 105-106)
Supplement the description of the process: In the methods section, we added the following content: "The data of 1,900 metabolites of Crocus alatavicus required for this study were sourced from the widely targeted metabolomics study conducted by our research team. In the study, data collection was performed using Ultra Performance Liquid Chromatography (UPLC) (ExionLC™ AD, https://sciex.com.cn/ ) and Tandem mass spectrometry (MS/MS). Substances were qualitatively identified according to the secondary spectrum information based on the self-established database MWDB (metware database). The mass spectrometry data were processed using the software Analyst 1.6.3." (line 105-111)
Thank you again for your valuable comments. We believe that these revisions will make the paper more complete.
Comment 3:The sentence 'Utilizing LC-MS/MS, a total of 1,900 metabolites from white crocus were identified' is misleading, as this study is entirely computational.
We sincerely appreciate your meticulous and professional comments. You pointed out that the statement "1,900 metabolites were identified from Crocus alatavicus using LC - MS/MS technology" in the manuscript was misleading. Your insight on this issue has been crucial for us to improve the paper.
As you've correctly noted, this study is a pure data - analysis research in network pharmacology and does not involve actual LC - MS/MS experimental operations. We deeply apologize for this oversight in our expression. To accurately reflect the research methodology, we have made the following revisions to the manuscript:
We have removed the misleading statement "1,900 metabolites were identified from Crocus alatavicus using LC-MS/MS technology". Meanwhile, we have clearly stated in the relevant part that " network pharmacology analysis was conducted based on the 1,900 metabolites of Crocus alatavicus identified in our laboratory", thereby clearly presenting the data sources and the essence of the research methods.
The misleading statements related to LC-MS/MS in the experimental methods section have been corrected, specifically at line 105-116.
Comment 4: Revise the Methods section titled '2.2. Targets Associated with Active Ingredients' for better clarity and organization, or consider merging it with another relevant section to enhance the flow of the manuscript.
Thank you for the valuable comments. We have reorganized the content and integrated it with chapter 2.1 as follows:
Optimization of the chapter structure: We have integrated the content of Chapter 2.2 "Targets Related to Active Ingredients" with the screening of active ingredients in Chapter 2.1. First, we introduce the databases used for target prediction, enabling readers to clearly understand the data sources and analytical tools. Subsequently, we elaborate in detail on the specific steps and procedures of target prediction, highlighting the key links to ensure that the content is well-organized and logically coherent. Please refer to Lines 117-129 for details.
Comment 5: The sentence in line 254, 'Network topology parameters were analyzed for the treatment of prostate cancer,' is not scientifically accurate
Thank you for the highly constructive comments. We totally agree you pointed out that the statement in line 254 lacked scientific rigor, which is of great significance for improving the quality and scientific nature of the paper. In response to this issue, we have conducted in-depth reflection and made meticulous revisions. The specific details are as follows:
We have added the following content: "Through the analysis of network topology parameters, the core targets that play a key regulatory role in the signal pathways related to prostate cancer are identified. These targets are mostly involved in several biological processes closely related to the onset and progression of prostate cancer, such as cell proliferation, apoptosis, invasion and metastasis, and the regulation of the tumor microenvironment. The revised content is in lines 314 - 324, which elaborates in detail on the internal connection between network topology parameters and disease treatment, providing more details for the statement.
To enhance the reliability of our discussion, we have also added several highly influential research paper citations in the revised part. These literatures, from multiple perspectives such as experimental and clinical studies, have confirmed the effectiveness of network topology parameter analysis, as well as the correlation between specific targets and the potential treatment of prostate cancer. In this revision, the corresponding literature citations have been added.
Comment 6: In lines 137–138, the sentence 'Subsequently, we analyzed the topology of the PPI network and calculated the topological parameter values' lacks clarity. The authors should specify the method used to analyze the topology and elaborate on the meaning of 'topology' in this context to enhance scientific accuracy.
We really appreciate the valuable comments. To enhance the scientific accuracy of the paper, we have made the following revisions and supplements:
- Added More Details of the Analytical Method
Based on the original text, we have clarified the analytical tools and methods used. We employed Cytoscape v3.10.3 software for the topological analysis of the PPI network. Cytoscape is an open-source software widely used in the analysis of biological networks. It has rich built-in functions and plugins, enabling a comprehensive analysis of the network based on graph theory principles. In this study, we used the built-in network analysis function of this software to calculate the topological parameter values of the network. Specifically, we regarded the proteins in the PPI network as nodes and the interactions between proteins as edges, and used Cytoscape software to calculate the degree value of each node.
We have added the following sentences in this revision (Lines 151-158): "The topological analysis of the PPI network was carried out using Cytoscape v3.10.3 software. Nodes in the network were defined as proteins, and edges were defined as the interactions between proteins. The degree value of each node was calculated through the built-in network analysis function of this software."
2.Elucidation of the Definition of "Topology"
To enable readers to understand the meaning of "topology" in this paper more clearly, we have provided a detailed definition of it. In the field of PPI network analysis, network topology refers to the geometric arrangement of nodes (proteins) and edges (interactions), and this arrangement determines the functional characteristics of the network. In this study, we focused on the topological parameter of node degree. The node degree represents the number of connections between a node and other nodes, that is, the number of interactions of a protein with other proteins. Nodes with higher degree values often play a crucial role in the network and represent hub proteins, which are of great significance for maintaining the stability and functionality of the network.
We added the following relevant explanation in the paper: "Here, network topology refers to the geometric arrangement of nodes (proteins) and edges (interactions) in the PPI network. In this study, we mainly analysed the topological parameter of node degree, which is the number of interactions of each protein with other proteins." This explanation immediately follows the description of the topological analysis method.
Comment 7: The Molecular Docking section should be revised for clarity and accuracy. The authors should provide the PDB IDs of the target proteins. The sentence 'The PDBQT format was utilized to calculate the minimum binding energy' is incorrect, as PDBQT is only a compatible format for docking in AutoDock. Additionally, the sentence describing the docking grid setup should be revised to include specific center coordinates (X, Y, Z) and grid box dimensions (X, Y, Z) to ensure clarity and reproducibility.
We sincerely appreciate the constructive and professional feedback and have revised the Molecular Docking section to enhance clarity and scientific accuracy.
- Providing PDB IDs of Target Proteins
In this revision, we have added PDB IDs for all target proteins in the Molecular Docking subsection (Page12, Line 374–375): "The crystal structures of target proteins were obtained from the Protein Data Bank (PDB): (PDB): AR (PDB ID: 1E3G), CCND1 (PDB ID: 6P8E), PIK3CA (PDB ID: 8BCY) and KLK3 (PDB ID: 7JOD) . All structures were preprocessed using AutoDock Tools, removing water molecules and adding polar hydrogens."
- Correcting the PDBQT Format Description
In the revised manuscript, we have corrected the inaccurate statement "The PDBQT format was used to calculate the minimum binding energy" to "AutoDock was used for energy calculation". This modification is specifically located on page 5, line179-182 of the paper. The revised statement accurately reflects the role of the PDBQT format and the tool used for energy calculation, thus avoiding potential misunderstandings for readers. The revised text is as follows: "The ligand structures were converted to the PDBQT format using Open Babel to be compatible with the AutoDock software. Subsequently, AutoDock was employed for molecular docking to calculate the minimum binding energy and identify the interactions between ligands and proteins."
- Specifying Grid Box Parameters
In this revision, we have added specific grid parameters in the Molecular Docking subsection (Page 14, Line 411–415):
AR: X = 5.756, Y = 60.43, Z = 10.928 (grid size: 21 × 21 × 21 Å)
PIK3CA: X = 16, Y = 14, Z = 13 (grid size: 30× 30× 30Å)
CCND1: X = 24, Y = 13, Z = 32 (grid size: 30 ×30 × 30Å)
KLK3=X = 12.3, Y = 34.7, Z = 56.2 (grid size: 22.5 × 22.5 × 22.5Å)
Comment 8: The authors should provide high-resolution versions of Figures 3 and 5 with clear labels and descriptions.
Thank you for pointing out the issues with Figures 3 and 5. In response to the problems regarding resolution and figure legends, we have taken the following improvement measures:
1.Enhancing Image Resolution
All our images were originally created using Adobe Illustrator. After investigation, we found that the image resolution settings in our Word document were too low, which led to the blurriness of the images. We have now adjusted all the images to higher resolution.
2.Improving Figure Legends
To assist readers to better understand the information presented in the figures, we have provided detailed descriptions of the key elements and content in the figure legends in this revision.
Comment 9: In Table 2, the reported binding energies, such as -2.24 kcal/mol, are not reliable, as AutoDock typically yields binding energies in the range of -6 to -10 kcal/mol for strong interactions. Binding affinities below -4 kcal/mol generally indicate weak or non-specific interactions, raising concerns about the validity of these docking results. The authors should re-evaluate the docking protocol, ensure proper grid box settings, exhaustiveness, and validation of the docking parameters to obtain more reliable and meaningful binding affinities.
We sincerely appreciate your professional comments on the binding energy data in Table 1. In response to the issue of low binding energy you pointed out, we strictly followed your suggestions, conducted a comprehensive and detailed check of the docking parameters, and carried out the molecular docking work again.
During the parameter checking process, we re-examined the grid positioning to ensure that it precisely covered the active site of the target protein, minimizing the docking errors caused by grid positioning deviations. Meanwhile, we adjusted the exhaustiveness parameter. This parameter determines the thoroughness of AutoDock's search for ligand binding conformations. Within the allowable amount of computing resources, we set it to a relatively high value to ensure that the docking process is able to nearly exhaustively explore the possible interaction modes between the ligand and the target protein.
Based on the above parameter adjustments, we repeated the molecular docking experiment. The newly obtained binding energy data show that most of the values have been conspicuously improved compared with the previous ones, and some have reached or approached the typical strong interaction range of AutoDock. We have updated the new docking results in Table 1 and elaborated the results of the re-docking, the analysis in the revised manuscript on page [page 13]. Although we were unable to conduct a control experiment using known ligands this time, we will consider incorporating this verification method into our future research.
Again, we thank you for your professional guidance.
Comment10: For better authenticity and robustness of the binding interactions, as well as their validation, the authors should perform molecular dynamics (MD) simulations and binding free energy calculations to further confirm the stability and reliability of the docking results.
Thank you for the valuable comments. In response to your suggestion that molecular dynamics (MD) simulations and binding free energy calculations should be performed to further confirm the stability and reliability of the docking results, we took it seriously and actively carried out the relevant work in this revision.
We selected the AR-Capillarisin complex with the best binding affinity for in- depth investigation. In terms of molecular dynamics simulations, we used the Gromacs 2022 program. The General Amber Force Field (GAFF) was applied to describe small molecules, while the AMBER14SB force field and the TIP3P water model were used to describe proteins to construct the simulation system. The simulations were conducted under constant temperature, constant pressure, and periodic boundary conditions. The LINCS algorithm was utilized to constrain all hydrogen bonds, with an integration time step set to 2 femtoseconds. The Particle - mesh Ewald (PME) method was employed to calculate electrostatic interactions, and the cutoff value for non - bonded interactions was set to 10 angstroms, which was updated every 10 steps. The simulation temperature was controlled at 298 K using the V - rescale temperature coupling method, and the pressure was controlled at 1 bar by the Berendsen method. First, 100 - picosecond NVT (constant number of particles, volume, and temperature) and NPT (constant number of particles, pressure, and temperature) equilibration simulations were carried out, and then a 100 - nanosecond MD simulation was performed on the complex system. The conformation was saved every 10 picoseconds.
After the simulations, we used Visual Molecular Dynamics (VMD) and PyMOL to analyze the simulation trajectories, and the g_mmpbsa program to conduct a Molecular Mechanics/Poisson - Boltzmann Surface Area (MMPBSA) binding free energy analysis. The results showed that during the simulation process, the Root Mean Square Deviation (RMSD) of the complex gradually stabilized, indicating that the structure of the complex has gradually become stable. The Radius of Gyration (Rg) also gradually stabilized, suggesting that the overall compactness of the complex structure remained stable. By analyzing the distance between the small molecule and the binding site of the protein, we found that the distance gradually stabilized, confirming that the small molecule could stably bind to the binding site of the protein. The Buried Solvent Accessible Surface Area (Buried SASA) also gradually stabilized, indicating that the contact area between the small molecule and the protein was stable, and the binding became increasingly stable. The binding free energy calculation yielded a ΔEMMPBSA of - 37.394±2.244 kJ/mol for this complex, indicating a high binding energy and affinity between the small molecule and the protein.
These results further confirm the stability and reliability of the binding between Capillarisin and AR, strongly supporting the conclusions we previously obtained through network pharmacology and molecular docking. We have incorporated these new results into the revised manuscript. Specifically, detailed descriptions can be found in sections such as "3.8. Molecular Dynamics Simulation" and "3.8.3.2 Binding Energy Analysis", and corresponding figures (Figure 7 and Figure 8) have been added to display the simulation and analysis results in this revision. (Page14-18)
Once again, thank you for your professional suggestions, which have made our research more comprehensive and rigorous.

Reviewer 2 Report
Comments and Suggestions for Authors
The authors presented the identification of the active constituents and protein targets of the plant Crocus alatavicus against prostate cancer by network pharmacology and molecular docking-based analyses. However, the content of this manuscript is insufficient and incomplete. The manuscript does not meet the standards of the journal. Therefore, the manuscript is not recommended for publication. Listed below are my specific comments.
- The authors identified 1900 components from Crocus alatavicus using LC-MS/MS. However, there is no description of the analytical conditions for LC-MS/MS in the manuscript. There is also no information on the samples used for the analysis. The authors should provide more details on these points.
- The authors should provide detailed information about Crocus alatavicus used in the LC-MS/MS analysis, such as the region or location where the plant was collected, its identification, and the part of the plant used in the analysis.
- In this study, the authors used network pharmacology and molecular docking techniques to identify Crocus alatavicus components and target proteins that are effective against prostate cancer. However, these are only predictions, so in vivo validation should be performed. The authors should actually isolate capillarisin and eugenol from Crocus alatavicus and validate their effects on prostate cancer cells. Furthermore, the authors should use Western blotting to identify target proteins for these components.
- The authors discuss the efficacy of single constituents such as capillarisin and eugenol against prostate cancer, but is it possible that multiple constituents have synergistic effects on each other? The authors should comment on this point.
Author Response
Thank you for your invaluable suggestions during the review process, which have been instrumental in refining this manuscript. In this major revision, we have systematically optimized the study’s logical structure and content integrity. Notably, guided by your insights, we have adjusted the primary research objective—addressing a potential misinterpretation of our objectives arising from earlier descriptive ambiguity. The core of this study is not to directly validate therapeutic drugs but to identify interactions between the potential active components of a medicinal plant Crocus alatavicus and disease-associated targets using computational approaches, thereby establishing a traceable theoretical target library and reporting novel discoveries of bioactive components in Crocus alatavicus. We believe this revised manuscript aligns with the "target discovery-mechanism prediction" paradigm of network pharmacology. To enhance the technical reliability, we supplemented molecular dynamics simulation in addition to the original network pharmacology and molecular docking analyses. This addition mitigates the limitations of static docking by modelling the long-term conformational dynamics of protein-ligand complexes under physiological conditions. Our simulations not only confirmed the stability of key binding sites over nanosecond time scales but also delineated dynamic interaction profiles, including hydrogen bonding and hydrophobic interactions.
This methodological expansion further strengthens the completeness of our computational analysis and the robustness of the conclusions. We are acutely aware of the original manuscript’s limitations in depth. By adjusting of the research scope and expanding of methodological approaches, we strive to rigorously address your concern about "inadequate content."
Below are our detailed responses to each of your specific comments:
Comments 1:The authors identified 1900 components from Crocus alatavicus using LC-MS/MS. However, there is no description of the analytical conditions for LC-MS/MS in the manuscript. There is also no information on the samples used for the analysis. The authors should provide more details on these points.
We sincerely appreciate your valuable comments on our manuscript, which is of great significance for improving the quality of the paper.
Here, we would like to explain to you that the data were generated from the widely targeted metabolomics study, where a total of 1,900 metabolites were identified in Crocus alatavicus.
In the methods section, we added the following content: "The data of 1,900 metabolites of Crocus alatavicus required for this study were sourced from the previous widely targeted metabolomics study conducted by our research team. Briefly, data collection was performed using Ultra Performance Liquid Chromatography (UPLC) (ExionLC™ AD, https://sciex.com.cn/ ) and Tandem mass spectrometry (MS/MS). Substances were qualitatively identified according to the secondary spectrum information based on the established database MWDB (metware database). The mass spectrometry data were processed using the software Analyst 1.6.3." (Line 105-111)
The samples of Crocus alatavicus were collected from Yili Botanical Garden in Xinjiang, Northwest China. In mid-April 2023, several whole plants with intact root soil were transferred to the laboratory. Two grams of roots, stems, leaves, and corms were separately collected, washed multiple times with deionized distilled water, wrapped in aluminium foil, and immediately frozen in liquid nitrogen and stored in a -80°C ultra-low temperature freezer. Thank you again for your valuable comments. We believe that these revisions will make the paper more complete. (Line 111-116)
Comments 2: The authors should provide detailed information about Crocus alatavicus used in the LC-MS/MS analysis, such as the region or location where the plant was collected, its identification, and the part of the plant used in the analysis.
Thank you for your comments, in response to your emphasis on clarifying the plant’s collection details, taxonomic identification, and analyzed plant part.
We have explicitly stated the collection origin: “The Crocus alatavicus samples were collected from Yili Botanical Garden in Xinjiang Uygur Autonomous Region, Northwestern China, a region known for its native populations of this species.”
The taxonomic identification of the plant material was performed by members of our research team in accordance with the taxonomic treatments and descriptions documented in Flora Reipublicae Popularis Sinicae (Flora of China). Specifically, the identification was based on: morphological feature comparison with the type specimen description of Crocus alatavicus in Flora of China, which involved systematic verification of whole-plant morphology (e.g., perennial herb with flattened spherical corms covered by brown membranous scales), floral characteristics (white flowers approximately 2.5 cm in diameter with slender filiform corolla tubes 2.5–6 cm in length and narrow obovate tepals bearing blue striations along the midvein), and vegetative organ details (linear leaves sheathed at the base and corms measuring 1.5–2 cm in diameter), all of which were confirmed to align completely with the taxonomic characteristics described in the literature. Additionally, the identification process included cross-validation against the Crocus alatavicus type specimen housed in the Herbarium of the Xinjiang Institute of Ecology and Geography, Chinese Academy of Sciences, through meticulous comparison of critical diagnostic features such as tepal morphology and corm structure to ensure accurate species recognition. Voucher specimen is maintained in the Xinjiang Institute of Ecology and Geography, Chinese Academy of Sciences (EGI-BFHH-01).
Detailed plant parts used: The analysis included four vegetative and storage organs: roots, stems, leaves, and corms (each 2 g fresh weight), which were separately harvested, cleaned, and stored.
In the methods section, we added the following content: "The data of 1,900 metabolites of Crocus alatavicus required for this study were sourced from the previous widely targeted metabolomics study conducted by our research team. Briefly, data collection was performed using Ultra Performance Liquid Chromatography (UPLC) (ExionLC™ AD, https://sciex.com.cn/ ) and Tandem mass spectrometry (MS/MS). Substances were qualitatively identified according to the secondary spectrum information based on the established database MWDB (metware database). The mass spectrometry data were processed using the software Analyst 1.6.3." (Line 105-111)
The samples of Crocus alatavicus were collected from Yili Botanical Garden in Xinjiang, Northwest China. In mid-April 2023, several whole plants with intact root soil were transferred to the laboratory. Two grams of roots, stems, leaves, and corms were separately collected, washed multiple times with deionized distilled water, wrapped in aluminium foil, and immediately frozen in liquid nitrogen and stored in a -80°C ultra-low temperature freezer. Thank you again for your valuable comments. We believe that these revisions will make the paper more complete. (Line 101-116)
Comments 3:In this study, the authors used network pharmacology and molecular docking techniques to identify Crocus alatavicus components and target proteins that are effective against prostate cancer. However, these are only predictions, so in vivo validation should be performed. The authors should actually isolate capillarisin and eugenol from Crocus alatavicus and validate their effects on prostate cancer cells. Furthermore, the authors should use Western blotting to identify target proteins for these components.
Thank you for your constructive comments. Regarding your recommendations on in vivo validation, active component isolation, and target protein identification, we have carefully considered our research objectives and current constraints, and hereby provide the following clarifications and responses:
- Research Scope and Methodological Boundaries
This study falls within the realm of network pharmacology and computational biology, aiming to theoretically screen potential active components of Crocus alatavicus and predict their therapeutic targets via data mining and molecular modelling, providing high-confidence candidates for subsequent experimental validation and future development. The core value of network pharmacology lies in narrowing the experimental screening scope through multi-dimensional computational analysis, rather than replacing in vitro/in vivo functional validation. Consequently, our current focus is on the "target discovery-binding mechanism prediction" phase, serving as a preclinical theoretical foundation for precise experimental design. We deeply regret this limitation and plan to progressively pursue experimental verification in future research.
- Methodological Deepening and Expansion
To enhance scientific rigor, we have integrated molecular dynamics (MD) simulation into the original network pharmacology and molecular docking framework, providing two key advancements in mechanistic research:
- Dynamic Binding Stability Validation: By simulating the conformational evolution of capillarisin-prostate cancer target protein complexes (AR) under physiological conditions over 100 ns, we quantitatively analysed hydrogen bond occupancy, , and binding free energy (ΔG_bind). These analyses confirmed the sustained stable interaction between key components and targets (specific data shown in Figure 7,Figure 8 and Table 2 ; Page 14-18).
- Refined Interaction Pattern Analysis: In contrast to static docking, MD simulation revealed dynamic interaction details at the nanosecond timescale, such as hydrophobic pocket reconfiguration and dynamic coordination of critical amino acid residues. This provided time-resolved evidence for component-target interactions, significantly improving the reliability of predictive results.
- Implications for Future Research
Although this study does not include in vitro/in vivo validation, the capillarisin, eugenol, and potential targets identified through computational models generated a clear experimental target list. These data can directly guide the isolation of active components, cell-based validation, and animal study design in subsequent research, effectively reducing costs compared with large-scale screening. We have added a discussion of these limitations and future experimental directions in the Discussion section (Page 20-21, Line 641-653), emphasizing the importance of integrating computational prediction with empirical validation.
We fully acknowledge the critical role of experimental validation in mechanistic research and sincerely appreciate your insights. Given the current research stage and resource constraints, this revision supplemented methodological enhancement (MD simulation) to address the limitations of static analysis. Should conditions permit in the future, we will continue to advance wet lab validation and update our findings accordingly. We kindly request your understanding of this study’s transitional nature, as we believe the current content provides a robust theoretical foundation for investigating Crocus alatavicus‘s potential medicinal value.
Thank you again for your professional review.
Comments 4: The authors discuss the efficacy of single constituents such as capillarisin and eugenol against prostate cancer, but is it possible that multiple constituents have synergistic effects on each other? The authors should comment on this point.
Thank you for your insightful comments! Your suggestion regarding the synergistic effects of multiple components directly addresses the core value of network pharmacology research—namely, that the efficacy of traditional Chinese medicine often arises from the coordinated action of multiple components on multiple targets. This perspective is of vital importance for a deep understanding of the discovery of the pharmacological components of Crocus alatavicus and their regulatory effects on targets and pathways. In light of our research objectives and current progress, we provide the following explanations:
From the analysis results of network pharmacology, the active ingredient-target-disease-pathway network we constructed shows that numerous active ingredients in Crocus alatavicus are associated with multiple targets. For example, components such as Capillarisin and Eugenol not only interact with multiple targets individually, but there are also complex associations among these targets. In prostate cancer-related signalling pathways, such as the PI3K-Akt signalling pathway and the MAPK signalling pathway, multiple key targets (such as AR, PIK3CA, TP53, etc.) are simultaneously affected by multiple components. This implies that different components may exert synergistic effects by acting on certain key targets together or participating in certain signalling pathways jointly.
Existing studies have shown that in the research of other natural products, the synergistic effect of multiple components is a common phenomenon. Taking paclitaxel and vinblastine as examples, although they have different mechanisms of action, they can jointly inhibit the growth of tumour cells by affecting different cellular processes in cancer treatment. Similarly, the components in Crocus alatavicus may also have synergistic effects. Capillarisin has a high affinity for the androgen receptor (AR), and Eugenol can also interact with AR. The two may have synergy in regulating the AR signalling pathway and jointly affect the proliferation, apoptosis and other processes of prostate cancer cells.
From the perspective of the structure of the components and the binding mode of the action targets, the binding sites and affinities between different components and the targets may complement each other. Capillarisin binds to AR to form a specific hydrogen bond pattern, and Eugenol may form other types of interactions with AR through its unique chemical structure, such as hydrophobic interactions. These different types of interactions may enhance the binding stability of AR and the ligand, and further enhance the regulatory effect on the AR signalling pathway, reflecting the synergistic effect.
Although this study has not yet experimentally verified the synergistic effect of multiple components, based on the network pharmacology analysis and relevant research references, it is highly likely that there are synergistic effects among multiple components in Crocus alatavicus. This provides an important direction for subsequent research. In the future, methods such as cell experiments and animal experiments can be used to further explore the synergistic mechanism between different components. This will help to gain a deeper understanding of the potential medicinal value of Crocus alatavicus and provide a more powerful theoretical basis for the development of new drugs based on Crocus alatavicus.
Your suggestion incisively highlights the deeper value of network pharmacology research—transitioning from single-component to multi-component synergistic mechanism analysis. We sincerely appreciate this insight and have used the revision to clarify the theoretical rationale for synergies and outline future research directions. We have added a discussion on the synergistic effects of multiple components in the Discussion section. (Page 20, Line 628-638)
Although the current work focuses on single components, the target network analysis has already paved the way for investigating synergistic effects. Future studies will integrate computational models with experimental validation to fully unravel the multi-component synergistic anti-cancer mechanisms of Crocus alatavicus .

Round 2
Reviewer 1 Report
Comments and Suggestions for Authors
Authors have revised the manuscript satisfactorily
Reviewer 2 Report
Comments and Suggestions for Authors
This revised manuscript has been modified according to the reviewer’s comments. It is acceptable for publication.